# NEURAL-SYMBOLIC RECURSIVE MACHINE FOR SYSTEMATIC GENERALIZATION

## ABSTRACT

Despite the tremendous success, existing machine learning models still fall short of human-like systematic generalization—learning compositional rules from limited data and applying them to unseen combinations in various domains. We propose Neural-Symbolic Recursive Machine (NSR) to tackle this deficiency. The core representation of NSR is a Grounded Symbol System (GSS) with combinatorial syntax and semantics, which entirely emerges from training data. NSR implements a modular design for neural perception, syntactic parsing, and semantic reasoning, which are jointly learned by a deduction-abduction algorithm. We prove that NSR is expressive enough to model various sequence-to-sequence tasks. Superior systematic generalization is achieved via the inductive biases of *equivariance* and *recursiveness* embedded in NSR. In experiments, NSR achieves state-of-the-art performance in three benchmarks from different domains: SCAN for semantic parsing, PCFG for string manipulation, and HINT for arithmetic reasoning. Specifically, NSR achieves 100% generalization accuracy on SCAN and PCFG and outperforms state-of-the-art models on HINT by about 23%. Our NSR demonstrates stronger generalization than pure neural networks due to its symbolic representation and inductive biases. NSR also demonstrates better transferability than existing neural-symbolic approaches due to less domain-specific knowledge required.

## 1 INTRODUCTION

A remarkable property underlying human intelligence is its *systematic compositionality*: the algebraic capacity to interpret an infinite number of novel combinations from finite known components (Chomsky, 1957)—"*infinite use of finite means*" (Chomsky, 1965). This type of compositionality is central to the human ability to generalize from limited data to novel combinations (Lake et al., 2017). Recently, several datasets have been proposed to test systematic generalization of machine learning models—SCAN (Lake & Baroni, 2018), PCFG (Hupkes et al., 2020), CFQ (Keysers et al., 2020), and HINT (Li et al., 2021), to name a few. While conventional neural networks fail dramatically on these datasets, certain inductive biases have been explored to improve systematic generalization. Csordás et al. (2021); Ontanón et al. (2022) improve Transformers' generalization performance by using relative positional encoding and sharing weights between layers. Chen et al. (2020) introduce a neural-symbolic stack machine to achieve nearly perfect accuracy on SCAN-like datasets. Despite the improved performance, these neural-symbolic methods often require domain-specific knowledge to design non-trivial symbolic components and are difficult to transfer to other domains.

To achieve human-like systematic generalization in a wide range of domains, we propose Neural-Symbolic Recursive Machine (NSR), which integrates the *joint* learning of perception, syntax, and semantics in a principled framework. The core representation of NSR is a Grounded Symbol System (GSS) (see Fig. 1), which entirely emerges from training data without domain-specific knowledge. NSR implements a modular design for neural perception, syntactic parsing, and semantic reasoning. Specifically, we first utilize a neural network as the perception module to ground symbols on the raw inputs. Next, the symbols are parsed into a syntax tree of the Grounded Symbol System by a transition-based neural dependency parser (Chen & Manning, 2014). Finally, we adopt *functional programs* to realize the semantic meaning of symbols (Ellis et al., 2021). Theoretically, we show that the proposed NSR is expressive enough to model various sequence-to-sequence tasks. Critically, the inductive biases of equivariance and recursiveness, encoded in each module, enable NSR to break

down the long input into small components, process them progressively, and compose the results, encouraging the model to learn meaningful symbols and their compositional rules. Such inductive biases are the crux of NSR's superb systematic generalization.

It is challenging to optimize NSR in an end-to-end fashion since annotations for the internal GSS are oftentimes unavailable and NSR is not fully differentiable. To tackle this issue, we present a probabilistic learning framework and derive a novel *deduction-abduction* algorithm to coordinate the joint learning of different modules. In the learning phase (see also Fig. 2), the model first performs greedy deduction over these modules to propose an initial GSS, which may yield wrong results. Next, a search-based abduction is applied top-down to search the neighborhood of initial GSS for possible solutions; such abduction revises the GSS until it generates the correct result. As a plausible solution, the revised GSS provides *pseudo* supervision to train each module, facilitating the learning of individual components in NSR.

We evaluate NSR on three benchmarks from various domains to study systematic generalization: (1) SCAN (Lake & Baroni, 2018), mapping natural language commands to action sequences; (2) PCFG (Hupkes et al., 2020), predicting the output sequences of string manipulation commands; (3) HINT (Li et al., 2021), predicting the results of handwritten arithmetic expressions. All these datasets include multiple splits for evaluating different aspects of systematic generalization. NSR achieves state-of-the-art performance on all these benchmarks. Specifically, NSR obtains 100% generalization accuracy on SCAN and PCFG and improves the state-of-the-art accuracy on HINT by about 23%. Result analyses reveal that NSR possesses stronger generalization than pure neural networks due to its symbolic representation and inductive bias. It also demonstrates better transferability than existing neural-symbolic approaches due to less domain-specific knowledge required. We also evaluate NSR on a proof-of-concept machine translation task from Lake & Baroni (2018) and the results demonstrate the promise of applying NSR to realistic domains.

## 2 RELATED WORK

There has been an increasing interest in studying the systematic generalization of deep neural networks. Started by the SCAN dataset (Lake & Baroni, 2018), multiple benchmarks across various domains have been proposed, including semantic parsing (Keysers et al., 2020; Kim & Linzen, 2020), string manipulation (Hupkes et al., 2020), visual question answering (Bahdanau et al., 2019), grounded language understanding (Ruis et al., 2020), and mathematical reasoning (Saxton et al., 2018; Li et al., 2021). These datasets serve as the test bed for evaluating different aspects of generalization, including systematicity and productivity. A line of research has developed different techniques for these datasets by injecting various inductive biases into deep neural networks. We categorize previous approaches into three classes by how they inject the inductive bias:

**Architectural Prior** The first class of methods explores different architectures of deep neural networks for compositional generalization. Dessì & Baroni (2019) found that convolutional networks are significantly better than recurrent networks in the "jump" split of SCAN. Russin et al. (2019) improved the standard RNNs by learning separate modules for syntax and semantics. Gordon et al. (2019) proposed the equivariant seq2seq model by incorporating convolution operations into RNNs to achieve local equivariance over permutation symmetries of interest, which are provided beforehand. Csordás et al. (2021) and Ontanón et al. (2022) observed that relative position encoding and sharing weights across layers significantly improve the systematic generalization of Transformers.

**Data Augmentation** The second class of methods designs different schemes to generate auxiliary training data for encouraging compositional generalization. Andreas (2020) performed data augmentation by replacing fragments of training samples with other fragments from similar samples, and Akyürek et al. (2020) trained a generative model to recombine and resample training data. The meta sequence-to-sequence model (Lake, 2019) and the rule synthesizer (Nye et al., 2020) are trained with samples drawn from a meta-grammar with a format close to the SCAN grammar.

**Symbolic Scaffolding** The third class of methods bakes symbolic components into neural architectures for improving compositional generalization. Liu et al. (2020) connected a memory-augmented neural model with analytical expressions, simulating the reasoning process. Chen et al. (2020) integrated a symbolic stack machine into a seq2seq framework and learned a neural controller to operate the machine. Kim (2021) learned latent neural grammars for both the encoder

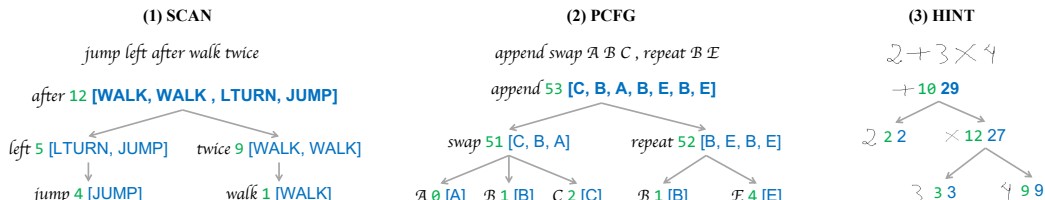

Figure 1: **Examples of Grounded Symbol Systems, which are interpretable and illustrate a human-like reasoning process.** (1) SCAN: each node is a triplet of (*word*, symbol, action sequence); (2) PCFG: each node is a triplet of (*word*, symbol, letter list). (3) HINT: each node is a triplet of (*image*, symbol, value). Each symbol is denoted by its index.

and the decoder in a seq2seq framework. These methods achieved strong generalization by injecting a symbolic scaffolding into their models. However, these symbolic components require domain knowledge for specialized modules and complicated training procedures, such as hierarchical reinforcement learning in Liu et al. (2020) and expensive search process for execution traces in Chen et al. (2020), which often have to be constrained under a carefully designed curriculum in practice. In contrast, the proposed NSR model requires less domain-specific knowledge for designing its modules and the proposed deduction-abduction algorithm for learning NSR does not require a specialized curriculum, leading to better transferability and easier optimization.

# 3 NEURAL-SYMBOLIC RECURSIVE MACHINE

## 3.1 REPRESENTATION: GROUNDED SYMBOL SYSTEM (GSS)

Two schools of thought, connectionism and symbolism, have a long debate about the proper representation of the human mind (Fodor et al., 1988; Fodor & Lepore, 2002; Marcus, 2018). Central to connectionism is *distributed representations* (Hinton, 1984), arguing that a certain concept or meaning is represented by a pattern of activity across many neurons. In contrast, symbolism postulates a *physical symbol system* (Newell, 1980), wherein each symbol alone represents an atomic concept, and more complicated concepts are formed by combining multiple symbols with certain syntax (Chomsky, 1965; Hauser et al., 2002; Evans & Levinson, 2009). Symbol systems are more interpretable and support stronger abstraction and generalization than distributed representations (Launchbury, 2017). However, handcrafting a symbol system for a domain requires a lot of domain-specific knowledge, and the constructed system is fragile and suffers from the notorious symbol grounding problem (Harnad, 1990).

In this work, we propose a grounded version of the symbol system as the internal representation for systematic generalization. Such a *Grounded Symbol System (GSS)* provides a principled integration of perception, syntax, and semantics, as exemplified by Fig. 1. Formally, a GSS is defined as a directed tree $T = < (x, s, v), e >$. Each node is a triplet of the grounded input $x$, the abstract symbol $s$, and the semantic meaning $v$. Each edge denotes the semantic dependency between the parent and the child node; an edge $i \rightarrow j$ denotes that the node $i$'s semantic meaning depends on node $j$'s.

Despite their nice properties, handcrafted symbol systems are inevitably fragile and labor-intensive. Therefore, it is essential to ground symbol systems on raw inputs and learn their syntax and semantics from the training examples, which we will discuss next.

## 3.2 MODEL: NEURAL-SYMBOLIC RECURSIVE MACHINE (NSR)

We now describe the proposed NSR that induces a GSS from training data. As illustrated in Fig. 2, the NSR consists of three learnable modules: A neural perception module for grounding symbols on the raw input, a dependency parser for predicting the dependencies between symbols, and a program synthesizer for predicting the semantic meanings. Since the ground-truth GSS is unavailable during training, these three modules ought to be learned in an end-to-end fashion without intermediate supervision. Below, we first describe the three modules of NSR and then discuss how to learn NSR end-to-end with our proposed deduction-abduction algorithm.

**Neural Perception** The role of the perception module is to map a raw input $x$ (*e.g.*, a handwritten expression) into a symbolic sequence $s$ (denoted by a list of indices). This perception module handles

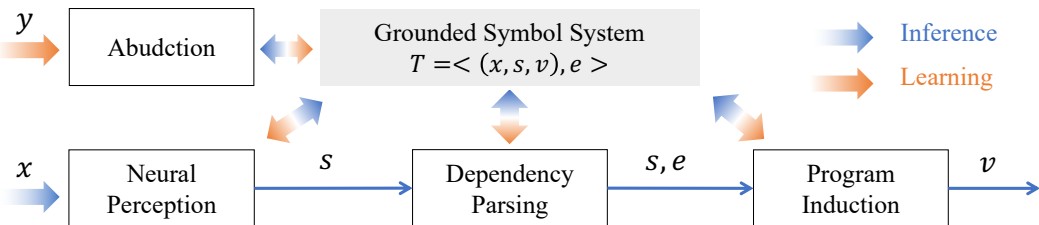

Figure 2: **The pipeline of inference and learning in NSR.**

the perceptual variance from raw input signals, such that each predicted token $w_i \in s$ is grounded in a certain part of the input $x_i \in x$. Formally, we have:

$$p(s|x; \theta_p) = \prod_i p(w_i|x_i; \theta_p) = \prod_i \text{softmax}(\phi(w_i, x_i; \theta_p)), \quad (1)$$

where $\phi(w_i, x_i; \theta_p)$ is a scoring function parameterized by a neural network with parameters $\theta_p$. The architecture of the perceptual neural network depends on the format of the raw input, which could be pre-trained; for example, we use a pre-trained convolutional neural network for images.

**Dependency Parsing**   To predict the dependencies between symbols, we adopt a transition-based neural dependency parser (Chen & Manning, 2014), commonly used for parsing natural language sentences. The transition-based dependency parser relies on a state machine that defines the possible transitions to parse the input sequence into a dependency tree. The parser constructs the parse tree from the input sequence by recursively applying the predicted transition to the state machine until the parsing process ends; see Fig. A1 for an illustration. At each step, the parser predicts one transition based on the state representation. The state representation is constructed from a local window and contains: (i) the top three words in the stack and buffer, (ii) the first and second leftmost/rightmost children of the top two words in the stack, and (iii) the leftmost of leftmost/rightmost of rightmost children of the top two words in the stack. Given the state representation, we adopt a two-layer feed-forward neural network to predict the transition. Formally, given the input sentence $s$, we have:

$$p(e|s; \theta_s) = p(\mathcal{T}|s; \theta_s) = \prod_{t_i \in \mathcal{T}} p(t_i|c_i; \theta_s), \quad (2)$$

where $\theta_s$ is the parser's parameters, $\mathcal{T} = \{t_1, t_2, ..., t_l\} \models e$ the transition sequence that generates the dependencies $e$, and $c_i$ the state representation at the $i$-th step. In practice, we use a greedy decoding strategy to predict the transition sequence for the input sentence.

**Program Induction**   Inspired by recent advances in program induction (Ellis et al., 2021; Balog et al., 2017; Devlin et al., 2017), we adopt *functional programs* to represent the semantics of symbols and view learning as program induction. Compared to purely statistical approaches, symbolic programs exhibit better generalizability and interpretability, and the learning is usually more sample-efficient. Formally, given the input symbols $s$ and their dependencies $e$, we have:

$$p(v|e, s; \theta_l) = \prod_i p(v_i|s_i, \text{children}(s_i); \theta_l) \quad (3)$$

where $\theta_l$ is the set of programs induced for every symbol. In practice, we use symbolic programs, yielding a deterministic reasoning process.

Learning semantics for a symbol is equivalent to searching for a program that matches the given examples. Candidate programs are composed of a set of pre-defined primitives. Based on the Peano axioms (Peano, 1889), we find a universal set of primitives: (1) `0`; (2) `inc`: $i \to i + 1$; (3) `dec`: $i \to i - 1$; (4)`==`; (5) `if`: $(a, b, c) \to$ if $a$ is true, return $b$, else $c$. Such a simple set of primitives are provably sufficient to represent any symbolic function, which we will further demonstrate in Sec. 3.4. To speed up the search process and facilitate generalization, we augment this set of primitives with a minimal subset of Lisp primitives for list processing (`cons`, `car`, `cdr`, *etc.*) and the recursion primitive (`Y`-combinator (Peyton Jones, 1987)). The `Y`-combinator allows to represent any recursive function and is the crux of extrapolation on semantics.

In practice, we adopt DreamCoder (Ellis et al., 2021) for program induction, which can efficiently synthesize programs from input-output pairs across a wide range of domains. We revised the original DreamCoder to handle noise in the given examples during the search process.

### 3.3 LEARNING

Since the intermediate GSS is latent and non-differentiable, the back-propagation is not applicable to learn NSR. Previous methods usually learn such neural-symbolic models with policy gradient algorithms, such as REINFORCE (Williams, 1992). However, the policy gradient methods have been shown to converge slowly or even fail to converge (Liang et al., 2018; Li et al., 2020); it requires generating a large number of samples over a large latent space with sparse rewards, hoping some samples may be lucky enough to hit high rewards for updating the policy. Due to the prohibitively large space of GSS, we have to seek for an alternative and more efficient learning algorithm.

Formally, let $x$ denote the input, $T = \langle (x, s, v), e \rangle$ the intermediate GSS, and $y$ the output. During learning, $(x, y)$ is observed but $T$ is latent. The likelihood of the observation $(x, y)$ marginalized over $T$ can be decomposed as:

$$p(y|x; \Theta) = \sum_T p(T, y|x; \Theta) = \sum_{s,e,v} p(s|x; \theta_p) p(e|s; \theta_s) p(v|s, e; \theta_l) p(y|v). \tag{4}$$

$s|x$ denotes the process of grounding symbols to raw signals, guided by the perception module $\theta_p$. $e|s$ denotes the process of parsing the symbol sequence into a parse tree, guided by the syntax module $\theta_s$. $v|s, e$ denotes the process of reasoning over the parse tree, guided by the semantic module $\theta_l$. $y|v$ is a deterministic process: $p(y|v) = 1$ if the final output of $v$ equals to $y$, otherwise 0.

From a maximum likelihood estimation (MLE) perspective, the learning objective is to maximize the observed-data log-likelihood $L(x, y) = \log p(y|x)$. Assuming $\theta_p, \theta_s, \theta_l$ are continuous, we take the derivative of $L$ w.r.t. $\theta_p, \theta_s, \theta_l$ (see Eq. (A2) for the detailed derivation):

$$\begin{aligned}
\nabla_{\theta_p} L(x, y) &= \mathbb{E}_{T \sim p(T|x,y)}[\nabla_{\theta_p} \log p(s|x; \theta_p)], \\
\nabla_{\theta_s} L(x, y) &= \mathbb{E}_{T \sim p(T|x,y)}[\nabla_{\theta_s} \log p(e|s; \theta_s)], \\
\nabla_{\theta_l} L(x, y) &= \mathbb{E}_{T \sim p(T|x,y)}[\nabla_{\theta_l} \log p(v|s, e; \theta_l)],
\end{aligned} \tag{5}$$

where $p(T|x, y)$ is the posterior distribution of $T$ given $(x, y)$. Since $p(y|v)$ can only be 0 or 1, $p(T|x, y)$ can be rewritten as:

$$p(T|x, y) = \frac{p(T, y|x; \Theta)}{\sum_{T'} p(T', y|x; \Theta)} = \begin{cases} 0, & \text{for } T \notin Q \\ \frac{p(T|x;\Theta)}{\sum_{T' \in Q} p(T'|x;\Theta)}, & \text{for } T \in Q \end{cases} \tag{6}$$

where $Q = \{T : p(y|T) = 1, T \in \Omega_T\}$ is the set of $T$ that match $y$. Usually, $Q$ is a very small subset of the entire space of $(s, e, v)$, *i.e.*, $Q \subseteq \Omega_T$.

Since taking expectation w.r.t. this posterior distribution (Eq. (5)) is intractable, we use Monte Carlo sampling to approximate it. The optimization procedure for an example $(x, y)$ is as follows:

1. Sample a plausible solution: $\hat{T} = \langle (x, \hat{s}, \hat{v}), \hat{e} \rangle \sim p(T|x, y)$.
2. Use $(x, \hat{s})$ to update the perception module ($\theta_p$).
3. Use $(\hat{s}, \hat{e})$ to update the syntax module ($\theta_s$).
4. Use $(\hat{s}, \hat{e}, \hat{v})$ to update the semantic module ($\theta_l$).

Steps 2–4 are supervised training for each module and no longer require the continuous assumption for the derivation of Eq. (5). Step 1 requires sampling from a highly sparse distribution in a large space, which we describe below.

**Deduction-Abduction** The key to the above learning procedure is the efficient sampling from the posterior distribution $p(T|x, y)$. To tackle this problem, we develop a novel *deduction-abduction* algorithm, summarized in Alg. A1. Concretely, for a given example $(x, y)$, we first perform greedy deduction from $x$ to obtain an initial GSS $T = \langle (x, \hat{s}, \hat{v}), \hat{e} \rangle$, which may yield a wrong result. To find a revised $T^*$ that matches the correct result $y$ during training, we search the neighbors of $T$ in a top-down manner by performing abduction over perception, syntax, and semantics; see Alg. A1 and Fig. A2 and A3. Theoretically, the above deduction-abduction process behaves as a Metropolis-Hastings sampler for the posterior distribution $p(T|x, y)$, as proven by Li et al. (2020).

### 3.4 EXPRESSIVENESS AND GENERALIZATION OF NSR

We now discuss NSR's properties. Overall, we find that NSR has the expressive power to model various seq2seq tasks, and the inductive biases in NSR enhance its systematic generalization capability.

**Expressiveness**   We prove that NSR has the expressive power to model various seq2seq tasks.

*Lemma* 1. For a finite unique set of $\{x^i : i = 0, ..., N\}$, there exists a neural network $f_p$ with enough capability, satisfying: $\forall x^i, f_p(x^i) = i$.

Provably, there exists a neural network that maps every element in a finite set to a unique index, *i.e.*, $x^i \rightarrow i$ (Hornik et al., 1989; Lu et al., 2017). The parsing process is trivial since every input is mapped to a single token.

*Lemma* 2. Any index space can be created from the primitives $\{0, \texttt{inc}\}$.

All indices are natural numbers, which can be recursively defined by $\{0, \texttt{inc}\}$: $0 \rightarrow 0, 1 \rightarrow \texttt{inc(0)}, 2 \rightarrow \texttt{inc(inc(0))}, 3 \rightarrow \texttt{inc(inc(inc(0)))}, ....$ We can create indices for both inputs and outputs via this lemma.

Equipped with the above lemmas, we have the following theorem on the expressiveness of NSR:

*Theorem* 3. For any finite dataset $D = \{(x^i, y^i) : i = 0, ..., N\}$, there exists an NSR that can sufficiently express $D$ using four primitives $\{0, \texttt{inc}, \texttt{==}, \texttt{if}\}$.

This theorem can be proven by constructing a degenerate NSR to "memorize" all examples in $D$ via a long program:

$$\text{NSR}(x) = \texttt{if}(f_p(x)\texttt{==}0, y^0, \texttt{if}(f_p(x)\texttt{==}1, y^1, ...\texttt{if}(f_p(x)\texttt{==}N, y^N, \varnothing)...) \tag{7}$$

The above program is akin to a lookup table, composed by the primitives $\{\texttt{if}, \texttt{==}\}$ and the index space created by the primitives $\{0, \texttt{inc}\}$. Since these four primitives are universal for all domains, we are assured that NSR is expressive enough to model various seq2seq tasks and thus enjoys better transferability than previous neural-symbolic approaches.

**Generalization**   Although the degenerate program defined by Eq. (7) achieves perfect accuracy on the training set, it can hardly generalize beyond training examples. To achieve strong generalization, we have to impose certain inductive biases; these inductive biases should be minimal and applicable to most domains. Inspired by previous progress on compositional generalization (Gordon et al., 2019; Chen et al., 2020), we hypothesize two necessary inductive biases: *equivariance* and *recursiveness*. Intuitively, equivariance enables the model's systematicity, generalizing from $\{$"run", "run twice", "jump"$\}$ to "jump twice", whereas recursiveness enables the model's productivity, generalizing to longer sequences like "run and jump twice".

Following Gordon et al. (2019), we formalize the hypothesis of equivariance and recursiveness from the perspective of group theory. A discrete group $\mathcal{G}$ is a set of elements $\{g_1, ..., g_{|\mathcal{G}|}\}$ with a binary group operation "·", satisfying the group axioms (closure, associativity, identity, and invertibility). For equivariance, we consider a *permutation* group $\mathcal{P}$, whose elements are permutations of a set $\mathcal{X}$. For $p \in \mathcal{P}$, we define the permutation operation $T_p : \mathcal{X} \rightarrow \mathcal{X}$. For recursiveness, we consider a *composition* operation $\mathcal{C}$ and define $T_c : (\mathcal{X}, \mathcal{X}) \rightarrow \mathcal{X}$. Formally, we have:

*Definition* 1 (Equivariance). Given two sets $\mathcal{X}$ and $\mathcal{Y}$, let $\mathcal{P}$ be a *permutation* group, whose group operations on $\mathcal{X}$ and $\mathcal{Y}$ are denoted by $T_p : \mathcal{X} \rightarrow \mathcal{X}$ and $T_p' : \mathcal{Y} \rightarrow \mathcal{Y}$, respectively. A mapping $\Phi : \mathcal{X} \rightarrow \mathcal{Y}$ is *equivariant* if and only if

$$\forall x \in \mathcal{X}, p \in \mathcal{P} : \Phi(T_p x) = T_p' \Phi(x).$$

*Definition* 2 (Recursiveness). Given two sets $\mathcal{X}$ and $\mathcal{Y}$, let $T_c : (\mathcal{X}, \mathcal{X}) \rightarrow \mathcal{X}$ and $T_c' : (\mathcal{Y}, \mathcal{Y}) \rightarrow \mathcal{Y}$ $\mathcal{C}$ be two *composition* operations on $\mathcal{X}$ and $\mathcal{Y}$, respectively. A mapping $\Phi : \mathcal{X} \rightarrow \mathcal{Y}$ is *recursive* if and only if

$$\exists T_c, T_c', \forall x_1 \in \mathcal{X}, x_2 \in \mathcal{X} : \Phi(T_c(x_1, x_2)) = T_c'(\Phi(x_1), \Phi(x_2)).$$

It is straightforward to see that the three modules of NSR, *i.e.*, neural perception (Eq. (1)), dependency parsing (Eq. (2)), and program induction (Eq. (3)) are equivariant and recursive, because they are all pointwise transformations according to their formulations. Empirically, models that achieve superb compositional generalization on certain reasoning tasks, including NeSS (Chen et al., 2020), LANE (Liu et al., 2020), and the proposed NSR, are equivariant and recursive. Therefore, we propose the following hypothesis about compositional generalization:

*Hypothesis* 1. If a model achieves compositional generalization, the mapping instantiated by the model $\Phi : \mathcal{X} \rightarrow \mathcal{Y}$ is *equivariant* and *recursive*.

Table 1: **The test accuracy on different splits of SCAN and PCFG.** "A. R." stands for "Around Right," "Sys." for "Systematicity," and "Prod." for "Productivity." The results of NeSS on PCFG are reported by adapting the source code provided by Chen et al. (2020) on PCFG.

| Model | SCAN | | | | PCFG | | |
|---|---|---|---|---|---|---|---|
| | Simple | Jump | A. R. | Length | IID | Sys. | Prod. |
| Seq2Seq (Lake & Baroni, 2018) | 99.7 | 1.2 | 2.5 | 13.8 | 79 | 53 | 30 |
| CNN (Dessì & Baroni, 2019) | 100.0 | 69.2 | 56.7 | 0.0 | 85 | 56 | 31 |
| Transformer (Csordás et al., 2021) | - | - | - | 20 | - | 96 | 85 |
| Transformer (Ontanón et al., 2022) | - | 0.0 | - | 19.6 | - | 83 | 63 |
| Equivariant Seq2seq (Gordon et al., 2019) | 100.0 | 99.1 | 92.0 | 15.9 | - | - | - |
| NeSS (Chen et al., 2020) | 100.0 | 100.0 | 100.0 | 100.0 | ~0 | ~0 | ~0 |
| NSR (ours) | 100.0 | 100.0 | 100.0 | 100.0 | 100 | 100 | 100 |

## 4 EXPERIMENTS

### 4.1 SCAN

The SCAN dataset (Lake & Baroni, 2018) has been widely used to evaluate the systematic generalization of machine learning models. The task aims to translate a natural language command into a sequence of actions, simulating navigation in a grid world.

**Evaluation** Similar to previous work (Lake, 2019; Gordon et al., 2019; Chen et al., 2020), we evaluate NSR in the following four splits. (1) Simple: randomly split samples for training and testing. (2) Length: the output sequences in the training set contain at most 22 actions, whereas the output lengths in the test set are between 24 and 48. (3) Jump: the primitive "jump" only appears alone in the training set, whereas the test set contains commands combining "jump" with other primitives. (4) Around right: the phrase "around right" is held out from the training set, but both "around" and "right" occurs in the training set separately, such as "around left" and "opposite right."

**Baselines** We compare NSR with the following baselines: (1) Seq2seq (Lake & Baroni, 2018); (2) CNN (Dessì & Baroni, 2019); (3) Transformer (Csordás et al., 2021; Ontanón et al., 2022); (4) Equivariant seq2seq (Gordon et al., 2019), which incorporates convolutional operations into recurrent neural networks to achieve local equivariance; and (5) NeSS (Chen et al., 2020), which integrates a symbolic stack machine into a seq2seq framework.

**Results** Tab. 1 summarizes the results. Both NSR and NeSS achieve 100% accuracy on splits requiring systematic generalization, while other models' best performance on the Length split is only 20%. This contrast demonstrates the superiority of symbolic components (*i.e.*, the symbolic stack machine in NeSS and the Grounded Symbol System in NSR) for systematic generalization.

While both NeSS and NSR achieve perfect generalization on SCAN, we would like to highlight some key differences. First, NeSS requires a considerable amount of domain-specific knowledge to design the stack machine's components, such as the stack operations and the category predictors; NeSS without category predictors drops to around 20% for 3 out of 5 runs. Second, training NeSS requires a manually defined curriculum with customized training procedures for the latent category predictors. In contrast, our proposed NSR embraces a modular design with little domain-specific knowledge and does not require any special training scheme like a pre-defined curriculum.

Fig. 3 visualizes the syntax and semantics learned by NSR from the SCAN Length split. The dependency parser of NSR, which is equivariant as discussed in Sec. 3.4, forms clear permutation equivalent groups in terms of syntax among the input words: {turn, walk, look, run, jump}, {left, right, opposite, around, twice, thrice}, {and, after}. Note that we do not provide any prior information about these groups—they entirely emerge from the training data, unlike providing equivariant groups beforehand (Gordon et al., 2019) or explicitly incorporating a category induction procedure from execution traces (Chen et al., 2020). In the learned programs, the program synthesizer of NSR creates an index space for the target language and finds the correct programs for representing the semantics of each source word.

### 4.2 PCFG

Next, we conduct experiments on the PCFG dataset (Hupkes et al., 2020), in which the model learns to predict the output of a command of string manipulations, *e.g.*, append swap F G H,

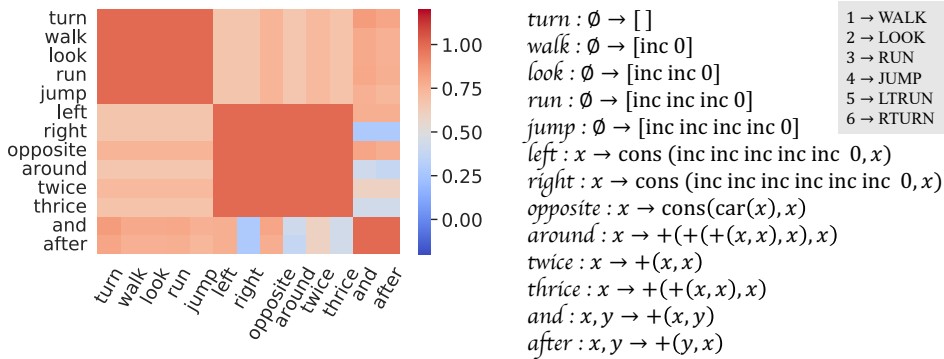

Figure 3: **Left**: the syntactic similarity between input words of NSR trained on the SCAN Length split. The similarity between word $i$ and word $j$ is measured by the percentage of test samples in which replacing $i$ with $j$, or vice versa, does not change the dependencies predicted by the dependency parser. **Right**: the induced programs for input words, where $x$ and $y$ denote the inputs, $\varnothing$ denotes empty inputs, `cons` inserts an item to the first position of a list, `car` returns the first item in a list, and $+$ concatenates two lists.

`repeat I J → H G F I J I J`. The input sequences in PCFG are generated by a probabilistic context-free grammar, and the output sequences are constructed by recursively applying the string edit operations that are specified in the input sequences. The input samples are selected to match the statistical properties of a textual corpus from natural languages, such as the lengths of the sentences and depths of the parse trees.

**Evaluation**   We evaluate models on the following splits. (1) IID: randomly split samples for training and testing. (2) Systematicity: this split focuses explicitly on models' ability to interpret pairs of functions that were never seen together during training. (3) Productivity: this split focuses on generalization to longer sequences; *i.e.*, the training samples contain up to 8 functions, while the test samples contain at least 9 functions. We compare with the following baselines: LSTM and CNN (Hupkes et al., 2020), Transformer (Csordás et al., 2021; Ontanón et al., 2022), and NeSS (Chen et al., 2020).

**Results**   Tab. 1 summarizes the results. NSR achieves 100% accuracy on all splits of PCFG and outperforms the previous state-of-the-art model (Transformer) by 4% on the "Systematicity" split and 15% on "Productivity." Notably, NeSS completely fails on the PCFG dataset, although it achieves perfect accuracy on SCAN. By inspecting the training process of NeSS on PCFG, we find that the stack operations of NeSS cannot represent the binary functions in PCFG, and the trace search process fails due to the large vocabulary and the long sequences of PCFG. Domain-specific knowledge and significant efforts would be required to re-design the stack machine and the training procedure.

## 4.3   HINT

Finally, we evaluate NSR on the HINT dataset (Li et al., 2021), in which the model learns to predict the integer result of a handwritten arithmetic expression without any intermediate supervision, *e.g.*, ( 3+2 )✗8 → 40. Compared with SCAN and PCFG, HINT is much more challenging since its perception deals with real handwritten images with high variance and ambiguity, its syntax is more complicated due to the existence of parentheses, and its semantics involves recursive functions. The HINT dataset comes with a single training set and five test subsets for evaluating different generalizations across perception, syntax, and semantics.

**Evaluation**   Following Li et al. (2021), we train models on the single training set and evaluate them on the following five test subsets. (1) "I": expressions are seen in the training set, but composed of unseen handwritten images. (2) "SS": expressions are unseen, but their lengths and values are within the same range as training. (3) "LS": expressions are longer than training, but their values are within the same range. (4) "SL": expressions have larger values and their lengths are the same as training. (5) "LL": expressions are longer, and their values are bigger than training. A prediction is considered correct if and only if it is exactly equal to the ground-truth result.

**Baselines**   We compare the proposed NSR with the following baselines: the seq2seq neural models (*i.e.*, GRU, LSTM, and Transformer) (Li et al., 2021) and NeSS (Chen et al., 2020). Note that each model is equipped with a ResNet-18 as the image encoder.

Table 2: **The test accuracy on HINT.** The results of GRU, LSTM, and Transformer are cited from Li et al. (2021), and the results of NeSS are reported by adapting its source code on HINT.

| Model | Symbol Input | | | | | | Image Input | | | | | |
|---|---|---|---|---|---|---|---|---|---|---|---|---|
| | I | SS | LS | SL | LL | Avg. | I | SS | LS | SL | LL | Avg. |
| GRU | 76.2 | 69.5 | 42.8 | 10.5 | 15.1 | 42.5 | 66.7 | 58.7 | 33.1 | 9.4 | 12.8 | 35.9 |
| LSTM | 92.9 | 90.9 | 74.9 | 12.1 | 24.3 | 58.9 | 83.9 | 79.7 | 62.0 | 11.2 | 21.0 | 51.5 |
| Transformer | 98.0 | 96.8 | 78.2 | 11.7 | 22.4 | 61.5 | 88.4 | 86.0 | 62.5 | 10.9 | 19.0 | 53.1 |
| NeSS | $\sim$0 | $\sim$0 | $\sim$0 | $\sim$0 | $\sim$0 | $\sim$0 | - | - | - | - | - | - |
| NSR (ours) | **98.0** | **97.3** | **83.7** | **95.9** | **77.6** | **90.1** | **88.5** | **86.2** | **67.1** | **83.2** | **58.2** | **76.0** |

Figure 4: **The evolution of learned programs in NSR for HINT.** The recursive programs in DreamCoder are represented by lambda calculus (*a.k.a.* $\lambda$-calculus) with Y-combinator. Here, we translate the induced programs into pseudo code for easier interpretation. Note that there might be different yet functionally-equivalent programs to represent the semantics of a symbol; we only visualize a plausible one here.

**Results** Tab. 2 summarizes the results on HINT. Comparing NSR with seq2seq neural models, we can see that NSR improves the state-of-the-art performance (Transformer) by about 23%. By inspecting the results on each test subset, we find that the improvement primarily comes from better extrapolation over syntax and semantics: NSR boosts the accuracy of "LL" from 19.0% to 58.2%, while the accuracy gains on "I" and "SS" are only around 2%. Please refer to Fig. A5 for qualitative examples of NSR's predictions on HINT. Again, NeSS completely fails on HINT due to similar reasons for its failure on PCFG as discussed in Sec. 4.2.

Fig. 4 illustrates the evolution of semantics along the training of NSR in HINT. This pattern is highly in accordance with how children learn arithmetic in developmental psychology (Carpenter et al., 1999): The model first masters the semantics of digits as *counting*, then learns + and − as recursive counting, and finally figures out how to define × and ÷ based on + and −. Crucially, × and ÷ are impossible to be correctly learned before mastering + and −. The model is endowed with such an incremental learning capability since the program induction module allows the semantics of concepts to be built compositionally from those learned earlier (Ellis et al., 2021).

## 5 CONCLUSION AND DISCUSSION

In this paper, we present NSR, which learns Grounded Symbol System from data to achieve systematic generalization. The Grounded Symbol System simulates a generalizable, interpretable representation and provides a principled integration of perception, syntax, and semantics. NSR adopts a modular design and encodes the inductive biases of equivariance and recursiveness in each module, which are critical for achieving compositional generalization. To train NSR without the supervision of GSS, we present a probabilistic learning framework and propose a novel deduction-abduction algorithm to facilitate the efficient learning of NSR. NSR achieves state-of-the-art performance on three benchmarks ranging from semantic parsing and string manipulation to arithmetic reasoning.

A proof-of-concept machine translation experiment demonstrates the promise of applying NSR to realistic domains. At the same time, we anticipate potential challenges of applying NSR to real-world tasks: (1) The noisy and numerous concepts in real-world tasks have a large space of grounded symbol system and might slow the training of NSR; (2) The functional programs in NSR are deterministic and thus not able to represent probabilistic semantics in real-world tasks, e.g., in machine translation, there might be multiple ways to translate a single sentence. How to solve these challenges will be explored in future work.

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

| ID | Stack Buffer | Transition | Dependency |
|----|--------------|------------|------------|
| 0 | $3 + 4 \times 2$ | Shift | |
| 1 | $3 + 4 \times 2$ | Shift | |
| 2 | $3 + 4 \times 2$ | Left-Arc | $3 \leftarrow +$ |
| 3 | $+ 4 \times 2$ | Shift | |
| 4 | $+ 4 \times 2$ | Shift | |
| 5 | $+ 4 \times 2$ | Left-Arc | $4 \leftarrow \times$ |
| 6 | $+ \times 2$ | Shift | |
| 7 | $+ \times 2$ | Right-Arc | $\times \rightarrow 2$ |
| 8 | $+ \times$ | Right-Arc | $+ \rightarrow \times$ |

Start: σ = [ROOT], β = $w_1, ..., w_n$, A = ∅
1. Shift  σ, $w_i$|β, A ➜ σ|$w_i$, β, A
2. Left-Arc$_r$  σ|$w_i$|$w_j$, β, A ➜ σ|$w_j$, β, A∪{$r(w_j,w_i)$}
3. Right-Arc$_r$  σ|$w_i$|$w_j$, β, A ➜ σ|$w_i$, β, A∪{$r(w_i,w_j)$}
Finish: σ = [$w$], β = ∅

$$3 + 4 \times 2$$

Figure A1: **Applying the transition-based dependency parser to an example of HINT.** It is similar for SCAN and PCFG.

# A APPENDIX

## A.1 MODEL DETAILS

**Dependency Parsing**  Fig. A1 illustrate the process of parsing an arithmetic expression via the dependency parser. Formally, a *state* $c = (\alpha, \beta, A)$ in the dependency parser consists of a *stack* $\alpha$, a *buffer* $\beta$, and a set of *dependency arcs* $A$. The initial state for a sequence $s = w_0 w_1 ... w_n$ is $\alpha = [\texttt{Root}], \beta = [w_0 w_1 ... w_n], A = \varnothing$. A state is regarded as terminal if the buffer is empty and the stack only contains the node $\texttt{Root}$. The parse tree can be derived from the dependency arcs $A$. Let $\alpha_i$ denote the $i$-th top element on the stack, and $\beta_i$ the $i$-th element on the buffer. The parser defines three types of transitions between states:

- LEFT-ARC: add an arc $\alpha_1 \rightarrow \alpha_2$ to $A$ and remove $\alpha_2$ from the stack $\alpha$. Precondition: $|\alpha| \geqslant 2$.
- RIGHT-ARC: add an arc $\alpha_2 \rightarrow \alpha_1$ to $A$ and remove $\alpha_1$ from the stack $\alpha$. Precondition: $|\alpha| \geqslant 2$.
- SHIFT: move $\beta_1$ from the buffer $\beta$ to the stack $\alpha$. Precondition: $|\beta| \geqslant 1$.

The goal of the parser is to predict a transition sequence from an initial state to a terminal state. The parser predicts one transition from $\mathcal{T} = \{\text{LEFT-ARC}, \text{RIGHT-ARC}, \text{SHIFT}\}$ at a time, based on the current state $c = (\alpha, \beta, A)$. The state representation is constructed from a local window and contains following three elements: (i) The top three words on the stack and buffer: $\alpha_i, \beta_i, i = 1, 2, 3$; (ii) The first and second leftmost/rightmost children of the top two words on the stack: $lc_1(\alpha_i), rc_1(\alpha_i), lc_2(\alpha_i), rc_2(\alpha_i), i = 1, 2$; (iii) The leftmost of leftmost/rightmost of rightmost children of the top two words on the stack: $lc_1(lc_1(\alpha_i)), rc_1(rc_1(\alpha_i)), i = 1, 2$. We use a special $\texttt{Null}$ token for non-existent elements. Each element in the state representation is embedded to a $d$-dimensional vector $e \in R^d$, and the full embedding matrix is denoted as $E \in R^{|\Sigma| \times d}$, where $\Sigma$ is the concept space. The embedding vectors for all elements in the state are concatenated as its representation: $c = [e_1 \, e_2 ... e_n] \in R^{nd}$. Given the state representation, we adopt a two-layer feed-forward neural network to predict the transition.

**Program Induction**  Program induction, *i.e.*, synthesizing programs from input-output examples, was one of the oldest theoretical frameworks for concept learning within artificial intelligence (Solomonoff, 1964). Recent advances in program induction focus on training neural networks to guide the program search (Kulkarni et al., 2015; Lake et al., 2015; Balog et al., 2017; Devlin et al., 2017; Ellis et al., 2018a;b). For example, Balog et al. (2017) train a neural network to predict properties of the program that generated the outputs from the given inputs and then use the neural network's predictions to augment search techniques from the programming languages community. Recently, Ellis et al. (2021) release a neural-guided program induction system, *DreamCoder*, which can efficiently discover interpretable, reusable, and generalizable programs across a wide range of domains, including both classic inductive programming tasks and creative tasks such as drawing pictures and building scenes. DreamCoder adopts a "wake-sleep" Bayesian learning algorithm to extend program space with new symbolic abstractions and train the neural network on imagined and replayed problems.

To learn the semantics of a symbol $c$ from a set of examples $D_c$ is to find a program $\rho_c$ composed from a set of primitives $L$, which maximizes the following objective:

$$\max_{\rho} p(\rho|D_c, L) \propto p(D_c|\rho)\, p(\rho|L), \tag{A1}$$

where $p(D_c|\rho)$ is the likelihood of the program $\rho$ matching $D_c$, and $p(\rho|L)$ is the prior of $\rho$ under the program space defined by the primitives $L$. Since finding a globally optimal program is usually intractable, the maximization in Eq. (A1) is approximated by a stochastic search process guided by a neural network, which is trained to approximate the posterior distribution $p(\rho|D_c, L)$. We refer the readers to DreamCoder (Ellis et al., 2021)[1] for more technical details.

## A.2 LEARNING

**Derivation of Eq. (5)**   Take the derivative of $L$ w.r.t. $\theta_p$,

$$
\begin{aligned}
\nabla_{\theta_p} L(x, y) &= \nabla_{\theta_p} \log p(y|x) = \frac{1}{p(y|x)} \nabla_{\theta_p} p(y|x) \\
&= \sum_T \frac{p(T, y|x; \Theta)}{\sum_{T'} p(T', y|x; \Theta)} \nabla_{\theta_p} \log p(s|x; \theta_p) \\
&= \mathbb{E}_{T \sim p(T|x,y)} [\nabla_{\theta_p} \log p(s|x; \theta_p)].
\end{aligned}
\tag{A2}
$$

Similarly, for $\theta_s, \theta_l$, we have

$$
\begin{aligned}
\nabla_{\theta_s} L(x, y) &= \mathbb{E}_{T \sim p(T|x,y)} [\nabla_{\theta_s} \log p(e|s; \theta_s)], \\
\nabla_{\theta_l} L(x, y) &= \mathbb{E}_{T \sim p(T|x,y)} [\nabla_{\theta_l} \log p(v|s, e; \theta_l)],
\end{aligned}
\tag{A3}
$$

**Deduction-Abduction**   Alg. A1 describes the procedure for learning NSR by the proposed deduction-abduction algorithm. Fig. A2 illustrate the one-step abduction over perception, syntax, and semantics in HINT and Fig. A3 visualizes a concrete example to illustrate the deduction-abduction process. It is similar for SCAN and PCFG.

## A.3 EXPERIMENTS

**Experimental Setup**   For tasks taking symbols as input (*i.e.*, SCAN and PCFG), the perception module is not required in NSR; For the task taking images as input, we adopt ResNet-18 as the perception module, which is pre-trained unsupervisedly (Van Gansbeke et al., 2020) on handwritten images from the training set. In the dependency parser, the token embeddings have a dimension of 50, the hidden dimension of the transition classifier is 200, and we use a dropout of 0.5. For the program induction, we adopts the default setting in DreamCoder Ellis et al. (2021). For learning NSR, both the ResNet-18 and the dependency parser are trained by the Adam optimizer (Kingma & Ba, 2015) with a learning rate of $10^{-4}$. NSR are trained for 100 epochs for all datasets.

**Qualitative Examples**   Fig. A4 and Fig. A5 show several examples of the NSR predictions on SCAN and HINT.

---

[1] https://github.com/ellisk42/ec

---

**Algorithm A1** Learning by Deduction-Abduction

---

1: **Input**: Training set $D = \{(x_i, y_i) : i = 1, 2, ..., N\}$
2: **Initial Module**: perception $\theta_p^{(0)}$, syntax $\theta_s^{(0)}$, semantics $\theta_l^{(0)}$
3: **for** $t \leftarrow 0$ to $T$ **do**
4:     Buffer $\mathcal{B} = \varnothing$
5:     **for** $(x, y) \in D$ **do**
6:         $T = \text{DEDUCE}(x, \theta_p^{(t)}, \theta_s^{(t)}, \theta_l^{(t)})$
7:         $T^* = \text{ABDUCE}(T, y)$
8:         $\mathcal{B} = \mathcal{B} \cup \{T^*\}$
9:     **end for**
10:    $\theta_p^{(t+1)}, \theta_s^{(t+1)}, \theta_l^{(t+1)} = \text{learn}(\mathcal{B}, \theta_p^{(t)}, \theta_s^{(t)}, \theta_l^{(t)})$
11: **end for**
12: **return** $\theta_p^{(T)}, \theta_s^{(T)}, \theta_l^{(T)}$

---

1: **function** DEDUCE$(x, \theta_p, \theta_s, \theta_l)$
2:     sample $\hat{s} \sim p(s|x; \theta_p), \hat{e} \sim p(e|\hat{s}; \theta_s), \hat{v} = f(\hat{s}, \hat{e}; \theta_l)$
3:     **return** $T = <(x, \hat{s}, \hat{v}), \hat{e}>$
4: **end function**

---

1: **function** ABDUCE$(T, y)$
2:     Q=PriorityQueue()
       Q.push(root$(T), y, 1.0$)
3:     **while** $A, y_A, p$ = Q.pop() **do**
4:       $A = (x, w, v, arcs)$
5:       **if** $A.v == y_A$ **then**
6:         **return** $T(A)$
7:       **end if**
8:                                                                     ▷ *Abduce perception*
9:       **for** $w' \in \Sigma$ **do**
10:        $A' = A(w \rightarrow w')$
11:        **if** $A'.v == y_A$ **then**
12:          Q.push($A', y_A, p(A')$)
13:        **end if**
14:       **end for**
15:                                                                      ▷ *Abduce syntax*
16:       **for** $arc \in arcs$ **do**
17:        $A' = \text{rotate}(A, arc)$
18:        **if** $A'.v == y_A$ **then**
19:          Q.push($A', y_A, p(A')$)
20:        **end if**
21:       **end for**
22:                                                               ▷ *Abduce semantics*
23:       $A' = A(v \rightarrow y_A)$
24:       Q.push($A', y_A, p(A')$)
25:                                                                 ▷ *Top-down search*
26:       **for** $B \in \text{children}(A)$ **do**
27:        $y_B = \text{SOLVE}(B, A, y_A | \theta_l(A.w))$
28:         Q.push($B, y_B, p(B)$)
29:       **end for**
30:     **end while**
31: **end function**

---

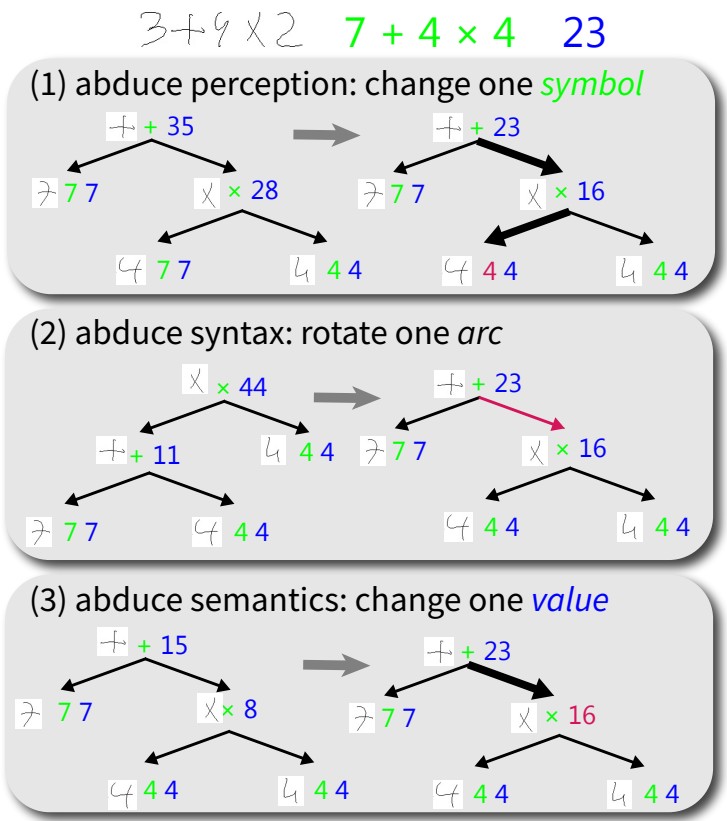

Figure A2: **Abduction over perception, syntax, and semantics in HINT.** Parts revised during abduction are highlighted in red.

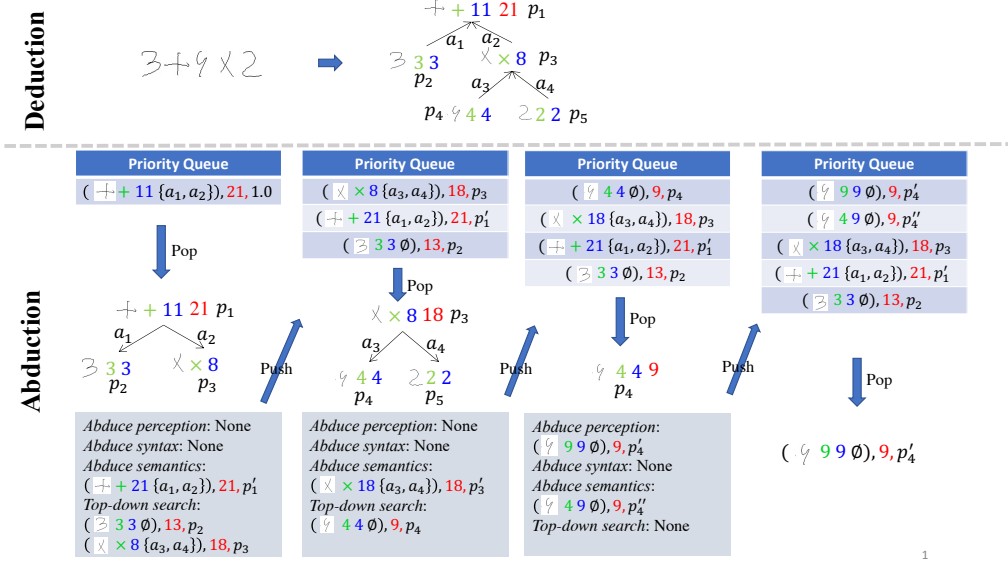

Figure A3: **An illustration of the deduction-abduction process for an example of HINT.** Given a handwritten expression, the system first performs a greedy deduction to propose an initial solution, which generates a wrong result. In abduction, the root node, paired with the ground-truth result, is first pushed to the priority queue. The abduction over perception, syntax, and semantics is performed on the popped node to generate possible revisions. A top-down search is also applied to propagate the expected value to its children. All possible revisions are then pushed into the priority queue. This process is repeated until we find the most likely revision for the initial solution.

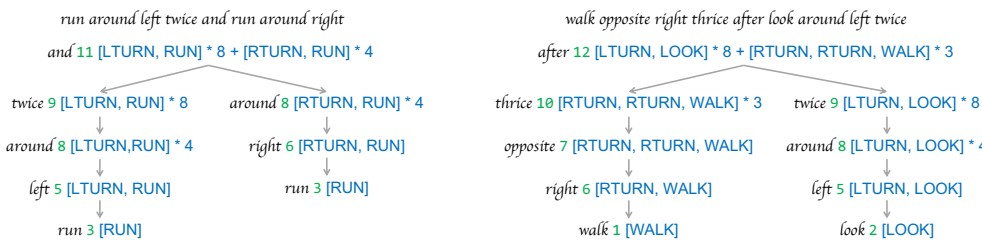

Figure A4: **Examples of NSR predictions on the test set of the SCAN Length split.** We use * (repeating the list) and + (concatenating two lists) to shorten the outputs for easier interpretation.

**Test subset I**

GT: (7+9/2)/3/8 = 1  PD: (7+9/2)/3/8 = 1

GT: 2/5-(0-1/6)/(8+2) = 1  PD: 2/5-(0-1/6(/(8+2) = 1

**Test subset SS**

GT: (3-1-(3-2))/(0+5) = 1  PD: (3-1-(3-2)/(0+5( = 1

GT: 3*(4-0+(6+(0*6-9))-6) = 12  PD: 3*(4-0+(6+(0*6-9))-6) = 24

**Test subset SL**

GT: 9*(9+8)*3-9/8 = 457  PD: 9*(9+8)*3-9/8 = 457

GT: (8*7*6+(3-0)/2*8)*7 = 2464  PD: (8*7*6+(3-0)/2*8)*7 = 448

**Test subset LS**

GT: (8/5+(1+5))*(4+5*0)-(7/(9*8)+1-3/(7+0)) = 31  PD: (8/5+(1+5)(*(4+5*0)-(7/(9*8)+1-3/(7+0() = 31

**Test subset LL**

GT: (8*7-5/5)*(3-(2-1)+1)/(9*1*(8+1)+(9+3)-0) = 2  PD: (8*7-5/5)*(3-(2-1)+1)/(9*1*(8+1)/(9+3)-0) = 24

Figure A5: **Examples of NSR predictions on the test set of HINT.** "GT" and "PD" denote "ground-truth" and "prediction," respectively. Each node in the tree is a tuple of (symbol, value).

## A.4  COMPOSITIONAL MACHINE TRANSLATION

To explore how well the proposed NSR model is applicable to real-world tasks, We run a proof-of-concept machine translation experiment using the English-French translation task from Lake & Baroni (2018), which is also used by previous works (Li et al., 2019; Chen et al., 2020; Kim, 2021) to explore their methods on realistic domains. Compared to synthetic tasks like SCAN and PCFG, this translation task contains more complex and ambiguous rules.

**Evaluation**   We adopt the public data splits from Li et al. (2019). Specifically, the training set contains 10,000 English-French sentence pairs, where the English sentences begin with phrases such as "I am", "you are", and their contractions. The training set also contains 1,000 repetitions of the sentence pair ("I am daxy", "je suis daxiste"), which is the only sentence that introduces the pseudoword "daxy" in the training set. The test set contains 8 combinations of "daxy" with other phrases, *e.g.*, "you are not daxy", which do not appear in the training set. Note that there are 2 different French translations of "you are" that frequently appear in the training set, so both of them are considered correct in the test set. We compare with the following baselines: Seq2Seq Lake & Baroni (2018), Primitive Substitution Li et al. (2019), and NeSS Chen et al. (2020).

**Results**   Tab. A1 summarizes the results of the compositional machine translation task. Similar to previous methods (Primitive Substitution and NeSS), NSR achieves a 100% generalization accuracy on this task. This experiment shows that NSR has the promise to be applied to real-world tasks. Despite the perfect accuracy in this proof-of-concept experiment, we indeed anticipate potential challenges of applying NSR to real-world tasks: (1) The noisy and numerous concepts in real-world tasks have a large space of grounded symbol system and might slow the training of NSR; (2) The functional programs in NSR are deterministic and thus not able to represent probabilistic semantics in real-world tasks, e.g., in machine translation, there might be multiple ways to translate a single sentence.

Table A1: **Accuracy on the compositional machine translation task.**

| Model | Accuracy |
|---|---|
| Seq2Seq | 12.5 |
| Primitive Substitution | 100.0 |
| NeSS | 100.0 |
| NSR (ours) | 100.0 |

## A.5  ABLATION STUDY ON HINT

To explore how well the individual modules of NSR are learned, we perform an ablation study on HINT to analyze the performance of each module of NSR. Specifically, along with the final results, the HINT dataset also provides the symbolic sequences and parse trees for evaluation. For Neural Perception, we report the accuracy of classifying each symbol. For Dependency parsing, we report the accuracy of attaching each symbol to its correct parent, given the ground-truth symbol sequence as the input. For Program Induction, we report the accuracy of final results, given the ground-truth symbol sequence and parse tree.

Table A2: **Accuracy of the individual modules of NSR on the HINT dataset.**

| Module | Neural Perception | Dependency Parsing | Program Induction |
|---|---|---|---|
| **Accuracy** | 93.51 | 88.10 | 98.47 |

Overall, each module achieves high accuracy, as shown in Tab. A2. For Neural Perception, most errors come from the two parentheses, "(" and ")", because they are visually similar. For Dependency Parsing, we analyze the parsing accuracies for different concept groups: digits (100%), operators (95.85%), and parentheses (64.28%). The parsing accuracy of parentheses is much lower than those of digits and operators. We think this is because, as long as digits and operators are correctly parsed in the parsing tree, where to attach the parentheses does not influence the final results because parentheses have no semantic meaning. For Program Induction, we can manually verify that the induced programs (Fig. 4) have correct semantics. The errors are caused by exceeding the recursion limit when calling the program for multiplication. The above analysis is also verified by the qualitative examples in Fig. A5.

