# OpenReview forum: "Neural-Symbolic Recursive Machine for Systematic Generalization"
_ICLR.cc/2023/Conference — Submitted to ICLR 2023_

### Official Review · Reviewer_oDdz · 2022-10-22

**Confidence:** 4
**Correctness:** 3
**Technical Novelty And Significance:** 3
**Empirical Novelty And Significance:** 2
**Recommendation:** 6

**Clarity, Quality, Novelty And Reproducibility:**

- the work is very clearly described, but with a few gaps. The deduction-abduction algorithm specifically applied here provides much of the novelty.
- With regards to reproducibility, standard tools like DreamCoder are used in some cases, and other approaches are fairly clear or very well-established (Sec 2.3). The appendices provide additional detail.

**Strength And Weaknesses:**

Strengths

- NSR is evaluated on three benchmarks from different domains, which demonstrates a good degree of generalization, and appears to achieve SotA on all three.
- The related work of Sec 4 is fairly comprehensive in topic, and clarity.
- The visualization of Fig 3 is fairly convincing

Weaknesses

- Although this is relatively minor, there are more recent alternatives to Chen & Manning (2014) for dependency parsing, including Mrini et al (2020) and a few based on deep biaffine attention.
- Also relatively minor, but it would be preferable to explore the space of program induction to a greater degree than in Sec 2.2, as this is a fairly active area of research.
- The baselines are adequate, but other more recent permutations of some of these exist (and it it probably not necessary to include seq2seq at this point). For example, is it possible to compare NSR against the approach in Ontañón et al (2022; arXiv:2108.04378v2), whom are cited?

Minor

- The forced correspondence to neuroscience always seems a bit naive, and may be removed? E.g., it’s easy to find studies that show that syntax and semantics are linked anatomically in a way that may counter Miller et al (2001), and connectionism vs symbolism is somewhat of a forced dichotomy.
- Please do another round of editing, as some grammatical errors exist throughout (e.g., “… and are thus difficult to transfer such capability …”)
- There should be a citation on the “have been shown” claim in Sec 2.3
- Additional detail should be added to the lemmas of Sec 2.4. E.g., the broad claim of the existence of a neural network in Lemma 1 is not in doubt but either a citation should be added to ‘provably’ or the abstracted proof should be added, as there is already a precedent for repeating previously established facts or approaches, from Sec 2.3
- It seems that ‘Theorem 3’ is meant to be ‘Lemma 3’ but, regardless, further explanation is advised.


**Summary Of The Paper:**

The paper introduces a ‘neural-symbolic recursive machine’ (NSR) to learn compositional rules from limited data that can be applied to unseen combinations in various domains. SotA is reported to be achieved on SCAN, PCFG, and HINT, which suggests at good domain-generalizability.

**Summary Of The Review:**


- This is a valuable extension of other work in this area, although it is only somewhat hampered by some vagueness, and potential strawmen of baselines, as opposed to direct comparisons with specific work.

---

> ### Author Response · Authors · 2022-11-18
> **Thank you for the constructive feedback!**
>
> We very much appreciate your efforts in reviewing our paper and providing insightful feedback. It is encouraging for us to see your positive comments that "NSR demonstrates a good degree of generalization," "the related work is fairly comprehensive in topic, and clarity," "Fig 3 is fairly convincing."
>
> We address your concerns as follows:
>
> > More recent alternatives to Chen & Manning (2014) for dependency parsing, including Mrini et al (2020).
>
> Thank you for the suggestion! When designing the individual modules for NSR, we intend to keep them as simple as possible and choose Chen & Manning (2014) for dependency parsing, which is powerful enough for the studied tasks in our experiments. Of note, NSR by design does not rely on the structure in Chen & Manning (2014); instead, it is compatible with more advanced alternatives for parsing like Mrini et al (2020), which is potentially helpful for applying NSR to domains with more complex syntax.
>
> > It would be preferable to explore the space of program induction to a greater degree
>
> Thank you for the suggestion! We will enrich the discussion of program induction in the revised draft as follows:
> "
> Program induction, i.e., synthesizing programs from input-output examples,
> was one of the oldest theoretical frameworks for concept learning within artificial intelligence (Solomonoff, 1964). Recent advances in program induction focus on training neural networks to guide the program search (Kulkarni et al., 2015; Lake et al., 2015; Balog et al., 2017; Devlin et al., 2017; Ellis et al., 2018a;b). For example, Balog et al. (2017) trained a neural network to first predict the program's properties given the input-output pairs, and then use the neural network's predictions to augment search techniques from the programming languages community. Recently, Ellis et al. (2021) devised a neural-guided program induction system, DreamCoder, which can efficiently discover interpretable, reusable, and generalizable programs across a wide range of domains, including both classic inductive programming tasks and creative tasks such as drawing pictures and building scenes. DreamCoder adopts a wake-sleep Bayesian learning algorithm to extend the program space with new symbolic abstractions and train the neural network on imagined and replayed problems.
> "
>
> > Compare NSR against the approach in Ontañón et al (2022)
>
> Thanks for pointing it out! Ontanón et al. (2022) evaluate the compositional generalization of Transformer variants and conduct experiments on SCAN and PCFG. We cite the best results from Ontanón et al. (2022) and compare them with the proposed NSR model as follows:
> |Model |SCAN-Jump |SCAN-Length |PCFG-Systematicity |PCFG-Productivity |
> |-------------------|:------------:|:------------:|:------------:|:------------:|
> | Transformer (Ontanón et al., 2022) | 0.0 | 19.6 | 82.8 | 63.4 |
> | NSR (ours) | 100.0 | 100.0 | 100.0 | 100.0 |
>
> From the above results, we can see that NSR demonstrates much stronger generalization than the state-of-the-art Transformer. We will add the above comparison in the revised draft.
>
> > Minor points.
>
> Thank you for the valuable suggestions! We will fix the mentioned issues in the revised draft.
>
> We hope the above response can resolve your questions and concerns. Please let us know if there is any further question!

---

### Official Review · Reviewer_j25h · 2022-10-24

**Confidence:** 3
**Correctness:** 3
**Technical Novelty And Significance:** 3
**Empirical Novelty And Significance:** 3
**Recommendation:** 6

**Clarity, Quality, Novelty And Reproducibility:**

The method section is difficult to follow, important figures and algorithms that are essential for understanding the algorithm are put in the appendix. This also makes the model difficult to reproduce.

**Strength And Weaknesses:**

Strength:
1. The proposed NSR achieve impressive generalization results on several datasets.
2. The three-module design is intuitive and general enough to be applied to different tasks and modalities.
3. Experiments on SCAN and PCFG show that NSR has good interpretability.

Weaknesses
1. The NSR is only tested on synthetic tasks. The limitation of applying NSR in real-world tasks is not discussed either.
2. The paper is difficult to follow, important diagrams and figures are put in the appendix.

**Summary Of The Paper:**

The paper proposes a new model -- Neural-Symbolic Recursive Machine (NSR), that uses a Grounded Symbol System as its core mechanism. The model includes three components: a perception module, a parsing module, and a program induction module. The three modules are trained in an end-to-end fashion without intermediate supervision. The system benefits from the inductive bias of equivariance and recursiveness. It shows a strong compositional generalization ability on several standard benchmarks, including SCAN, PCFG, and HINT.

**Summary Of The Review:**

Overall, the paper proposed an interesting new framework for achieving compositional generalization. Experiment results confirm the effectiveness of the model in synthetic tasks. However, the proposed method is fairly complicated, which shed doubt on whether the method can be generalized to real-world tasks. The author didn't provide any proof or discussion on this question.

---

> ### Author Response · Authors · 2022-11-18
> **Thank you for the constructive feedback!**
>
> We very much appreciate your efforts in reviewing our paper and providing insightful feedback. It is encouraging for us to see your positive comments that "NSR achieve(s) impressive generalization," "the three-module design is intuitive and general," and "NSR has good interpretability."
>
> We address your concerns as follows:
>
> > The NSR is only tested on synthetic tasks. The limitation of applying NSR in real-world tasks is not discussed either.
>
> To explore how well the proposed NSR model is applicable to real-world tasks, we run a proof-of-concept machine translation experiment using the English-French translation task from Lake & Baroni (2018), which is also used by previous works (Li et al., 2019; Chen et al., 2020; Kim, 2021) to explore their methods on realistic domains. Compared to synthetic tasks like SCAN and PCFG, this translation task contains more complex and ambiguous rules. To evaluate compositional generalization, for a word "dax," the training data contains only one pattern of sentence pairs ("I am daxy," "je suis daxiste"), but test data contains other patterns (e.g., "you are not daxy," "tu n es pas daxiste").
>
> We apply the proposed NSR model to this task. The results are as follows:
>
> |        Model        | Accuracy     |
> |-------------------|:------------:|
> |Seq2Seq (Lake & Baroni, 2018) | 12.5 |
> |Primitive Substitution (Li et al., 2019)| 100 |
> |NeSS (Chen et al., 2020)   | 100 |
> |NSR (ours)                 | 100 |
>
> Similar to previous methods (Primitive Substitution and NeSS), NSR achieves a 100\% generalization accuracy on this task. This experiment shows that NSR has the promise to be applied to real-world tasks. Despite the perfect accuracy in this proof-of-concept experiment, we indeed anticipate potential challenges of applying NSR to real-world tasks: (1) The noisy and numerous concepts in real-world tasks have a large space of grounded symbol system and might slow the training of NSR; (2) The functional programs in NSR are deterministic and thus not able to represent probabilistic semantics in real-world tasks, e.g., in machine translation, there might be multiple ways to translate a single sentence.
>
> We will add the above results, more details on this dataset, and the experimental setup into the revised draft.
>
>
> > The paper is difficult to follow, important diagrams and figures are put in the appendix. This also makes the model difficult to reproduce.
>
> Thanks for this valuable comment. We strive to make our paper easy to follow. Due to the page limit, we have to focus on the core idea and the intuition behind the model design in the main text, while putting **task-specific details** in the appendix. This way of organizing the paper is commented by other reviewers as "clear, well-structured and easy to follow", "clearly written and well organized", and "clearly described."
>
> We would love to further improve the clarity of our paper if you have a more specific suggestion on which part should move to the main text to make the method easier to follow. We are very happy to discuss and follow your suggestions!
>
> Regarding reproducibility, our code can be found at [bit.ly/nsr-iclr23](https://bit.ly/nsr-iclr23) and we will host the code in a GitHub repository and share the trained models and experimental logs in the future.
>
> We hope the above response can resolve your questions and concerns. Please let us know if there is any further question!

---

### Official Review · Reviewer_kopJ · 2022-10-27

**Confidence:** 3
**Correctness:** 3
**Technical Novelty And Significance:** 3
**Empirical Novelty And Significance:** 3
**Recommendation:** 6

**Clarity, Quality, Novelty And Reproducibility:**

The paper is clearly written and well organized. For a reader who is not very familiar with this field, it is not easy to tell how the proposed NSR is novel and different from existing work. This would be addressed by better contrasting NSR against prior work in the “Related Work” section. The proposed design of NSR using 3 modules to process perception, syntactic parsing and semantic reasoning to confer systemic generalization is intuitive and the experimental results show superior performance versus baselines. The proposed deduction-abduction algorithm to address the model’s training is sound and design choices are also well-explained.

**Strength And Weaknesses:**

Strength:
The proposed idea is sound and shows superior performance vs some baselines.
Theoretical analysis is done to show proposed NSR confers equivariance and recursiveness.

Weaknesses:
Lack of comparison versus more recent baselines such as Ontanón et al. (2022)


**Summary Of The Paper:**

The paper proposes Neural-Symbolic Recursive Machine (NSR) as a means for neural networks to learn compositional rules through 3 modules, one for processing perception, syntactic parsing and semantic reasoning. These modules made up the Grounded Symbol System (GSS) and are jointly learned through a deduction-abduction algorithm. The authors also showed that the NSR modules confer equivariance and recursiveness to the network’s representations. The experiments showed that NSR outperforms baselines on semantic parsing, string manipulation and arithmetic reasoning tasks.

**Summary Of The Review:**

Overall, the paper proposed a sound and intuitive approach for a model’s systemic generalization which is a timely contribution given how important generalization is for deep learning models. One area for improvement is more comparison against more recent baselines and prior work from the ‘data augmentation’ and ‘symbolic scaffolding’ categories. Though I am not an expert in this field, the contributions seem substantial enough for me to lean toward an accept.

---

> ### Author Response · Authors · 2022-11-18
> **Thank you for the constructive feedback!**
>
> We very much appreciate your efforts in reviewing our paper and providing insightful feedback. It is encouraging for us to see your positive comments that "the proposed idea is sound and shows superior performance," "the paper is clearly written and well organized," and "the proposed deduction-abduction algorithm is sound and design choices are also well-explained."
>
> We address your concerns as follows:
>
> > Lack of comparison versus more recent baselines such as Ontanón et al. (2022)
>
> Thanks for pointing it out! Ontanón et al. (2022) evaluate the compositional generalization of Transformer variants and conduct experiments on SCAN and PCFG. We cite the best results from Ontanón et al. (2022) and compare them with the proposed NSR model as follows:
> |Model |SCAN-Jump |SCAN-Length |PCFG-Systematicity |PCFG-Productivity |
> |-------------------|:------------:|:------------:|:------------:|:------------:|
> | Transformer (Ontanón et al., 2022) | 0.0 | 19.6 | 82.8 | 63.4 |
> | NSR (ours) | 100.0 | 100.0 | 100.0 | 100.0 |
>
> From the above results, we can see that NSR demonstrates much stronger generalization than the state-of-the-art Transformer. We will add the above comparison in the revised draft.
>
> > Contrast NSR against prior work in the "Related Work" section.
>
> Thank you for the suggestion! Our approach belongs to the third class ("Symbolic Scaffolding") discussed in Related Work; the most related method is the NeSS model proposed by Chen et al. (2020), which requires specific domain knowledge to design a symbolic stack machine with special operations (e.g., "CONCAT_M," "CONCAT_S"). Compared with NeSS, the proposed NSR model has two significant advantages:
> (1) The dependency parser in NSR adopts a standard stack without requiring any special stack operation beyond the standard ones (Shift, Left-Arc, Right-Arc).
> (2) The proposed deduction-abduction algorithm for learning NSR does not require a specialized curriculum for SCAN, as required by NeSS.
> Thus, NSR has better transferability than NeSS, as reflected by the experimental results (Table 1): NSR achieves 100% accuracy on both SCAN and PCFG, while NeSS succeeds on SCAN but fails on PCFG. We will make the comparison clearer and move the related work section to an earlier point in the revised draft.
>
> We hope the above response can resolve your questions and concerns. Please let us know if there is any further question!

---

### Official Review · Reviewer_w6P6 · 2022-11-05

**Confidence:** 4
**Correctness:** 4
**Technical Novelty And Significance:** 3
**Empirical Novelty And Significance:** 3
**Recommendation:** 5

**Clarity, Quality, Novelty And Reproducibility:**

**Clarity**: The writing of the paper’s abstract could be improved to increase the clarity, but the other sections’ writing quality is good. The multiple diagrams and visualizations  are clear and helpful to convey the ideas of the paper, and the key concepts (e.g. the analogy among different problems having underlying tree structure, and the NSR high-level pipeline). I recommend moving some of the visualizations in the appendix to the main paper.

**Novelty**: The proposed approach is new and the paper explores an important underexplored domain.

**Reproducibility**: Adding more details such as hyperparams and finer modeling decisions for the different component would be useful to increase the paper’s reproducibility.


**Strength And Weaknesses:**

**Strengths**:
* **Domain**: the paper explores an important and growing domain of systematic generalization.
* **Experiments and results**: The paper presents experiments in multiple complementary datasets covering parsing, semantic reasoning, compositional generalization and arithmetics. It achieves good results across multiple tasks, and compares to multiple suitable baselines. The descriptions in the experiments section of the tasks and the baselines are also clear, well-structured and easy to follow.
* **Modeling section**: The discussion in the modeling section is interesting and gives solid motivation for the approach presented in the paper, and the section is structured in a good and clear way. The model is also discussed from a theoretical perspective, and formally defines its properties (equivariance and recursiveness).
* **Related work, background and context**: the paper does a good job contextualizing the approach, providing the necessary general background, and covering the relevant related fields. I recommend moving the related work section to an earlier point in the paper (before the modeling section).

**Weaknesses**:
* **Related work comparison to specific recent approaches**: More details on how the new approach compares to recent models on the explored tasks would be useful.
* **Complicated approach**: The approach is a bit complicated and consists of multiple different modules that are responsible for different aspects of the problem. We know that models such as transformers do amazingly well on language tasks being more uniform, and so I recommend examining whether the model could be simplified without negatively and potentially also positively impacting its performance.
* **Undiferentaible**: The NSR model isn’t end-to-end differentiable, making training more challenging. Having multiple units that interact through hard predictions (without a gradient) could also lead to cascade errors, where early mistakes in early modules prevent the following ones to give the right predictions.I recommend exploring ways to make it differentiable, either by relaxing some of the constraints or using techniques such as gumbel-softmax etc.
* **Synthetic data**: The model is evaluated on synthetic data only. I would highly recommend exploring e.g. logical problems in natural language such as word problems, logical questions (e.g. LSAT), etc etc, to strengthen the paper.
* **Suggestion: focusing on the reasoning modules**: I also recommend exploring the parts of the model related to parsing and reasoning independently of the perceptual module since that is the central part of the paper, so would be good to highlight that, and also would broaden the range of datasets that could be explored.
* **Evaluating each component independently**: Providing more details on the performance of each component independently and performing ablation studies is critical for models with multiple components, rather than only having results on downstream task performance.

**Summary Of The Paper:**

The paper proposed a new approach composed of multiple components for systematic generalization on language tasks. It achieves good results on multiple datasets including SCAN, PCFG, and a task for arithmetic reasoning.

**Summary Of The Review:**

The paper explores an important direction and show nice experiments but I believe the main issue is the model that could be improved as well as some updates to the writing, and therefore overall at this point recommend weak rejection, but at the same time I wish to encourage the authors to work on improving the paper further to make it better!

---

> ### Author Response · Authors · 2022-11-18
> **(1/2) Thank you for the constructive feedback!**
>
> We very much appreciate your efforts in reviewing our paper and providing insightful feedback. It is encouraging for us to see your positive comments on the **Domain** ("important and growing"), **Experiments and results** ("good results across multiple tasks and compares to multiple suitable baselines", "The descriptions … are clear, well structured and easy to follow"), **Modeling section** ("The discussion is interesting and gives solid motivation"), and **Related work** ("does a good job contextualizing the approach …").
>
> We address your concerns as follows:
>
> > Related work comparison to specific recent approaches on the explored tasks
>
> Our approach belongs to the third class ("Symbolic Scaffolding") discussed in Related Work; the most related method is the NeSS model proposed by Chen et al. (2020), which requires specific domain knowledge to design a symbolic stack machine with special operations (e.g., "CONCAT_M," "CONCAT_S"). Compared with NeSS, the proposed NSR model has two significant advantages:
> (1) The dependency parser in NSR adopts a standard stack without requiring any special stack operation beyond the standard ones (Shift, Left-Arc, Right-Arc).
> (2) The proposed deduction-abduction algorithm for learning NSR does not require a specialized curriculum for SCAN, as required by NeSS.
>
> Thus, NSR has better transferability than NeSS, as reflected by the experimental results (Table 1): NSR achieves 100% accuracy on both SCAN and PCFG, while NeSS succeeds on SCAN but fails on PCFG. We will make the comparison clearer in the revised draft.
>
>
> > Complicated approach and undifferentiable.
>
> Being complicated and undifferentiable is a common weakness for most neural-symbolic approaches, including NeSS (Chen et al.2020), LANE (Liu et al. 2020), and the proposed NSR model, despite their better generalization than purely neural-based models like Transformer. We regard it **as an important future direction** to simplify these hybrid models and explore ways to make it differentiable.
>
> On the other hand, when designing NSR, we tried our best to simplify it and make it applicable for more domains, i.e., the dependency parser and the program induction are adopted in the most general form, without adding any domain-specific design. Although still undifferentiable, the proposed deduction-abduction algorithm makes the training more straightforward. It does not require a manually-designed curriculum, compared with previous neural-symbolic models like NeSS.
>
>
> > The model is evaluated on synthetic data only.
>
> To explore how well the proposed NSR model is applicable to real-world tasks, We run a proof-of-concept machine translation experiment using the English-French translation task from Lake & Baroni (2018), which is also used by previous works (Li et al., 2019; Chen et al., 2020; Kim, 2021) to explore their methods on realistic domains. Compared to synthetic tasks like SCAN and PCFG, this translation task contains more complex and ambiguous rules. To evaluate compositional generalization, for a word "dax," the training data contains only one pattern of sentence pairs ("I am daxy," "je suis daxiste"), but test data contains other patterns (e.g., "you are not daxy," "tu n es pas daxiste").
>
> We apply the proposed NSR model to this task. The results are as follows:
>
> |        Model        | Accuracy     |
> |-------------------|:------------:|
> |Seq2Seq (Lake & Baroni, 2018) | 12.5 |
> |Primitive Substitution (Li et al., 2019)| 100.0 |
> |NeSS (Chen et al., 2020)   | 100.0 |
> |NSR (ours)                 | 100.0 |
>
> Similar to previous methods (Primitive Substitution and NeSS), NSR achieves a 100\% generalization accuracy on this task. This experiment shows that NSR has the promise to be applied to real-world tasks. We will add the above results, more details on this dataset, and the experimental setup into the revised draft.

---

> > ### Author Response · Authors · 2022-11-18
> > **(2/2) Thank you for the constructive feedback!**
> >
> > > Suggestion: focusing on the reasoning modules
> >
> > Thank you for the suggestion! In the HINT experiment, we have already conducted experiments using symbol inputs (as shown by the columns of "Symbol Input" in **Table 2**), in which the model only has the parsing and the reasoning modules, without the perceptual module. We will emphasize these results and highlight them in the revised draft.
> >
> > > Evaluating each component independently
> >
> > Since only the HINT benchmark provides intermediate results for evaluation, we evaluate the three components of NSR individually on HINT by offering ground truth to other modules. The results are as follows:
> >
> > |        Module       | Accuracy     |
> > |-------------------|:------------:|
> > |Neural Peception|        93.51      |
> > |Dependency Parsing|      88.10      |
> > |Program Induction|       98.47      |
> >
> > Overall, each module achieves high accuracy. For Neural Perception, most errors come from the two parentheses, "(" and ")", because they are visually similar. For Dependency Parsing, we analyze the parsing accuracies for different concept groups: digits (100\%), operators (95.85\%), and parentheses (64.28\%). The parsing accuracy of parentheses is much lower than those of digits and operators. As long as digits and operators are correctly parsed in the parsing tree, where to attach the parentheses does not influence the final results because parentheses have no semantic meaning. For Program Induction, we can manually verify that the induced programs (Fig. 4) have correct semantics. The errors in the above table are caused by exceeding the recursion limit when calling the program for multiplication. We will add the above results and analysis in the revised draft.
> >
> > We hope the above response can resolve your questions and concerns. Please let us know if there is any further question!

---

### Author Response · Authors · 2022-11-18
**General comments for all reviewers and the summary of Revision**

Dear all reviewers,

We are grateful for your constructive comments and helpful feedback. We are very encouraged by the acknowledgment that our paper "explores an important and growing domain," (Reviewer w6P6) and "is clearly written and well organized" (Reviewers w6P6, kopJ, oDdz), the proposed idea is intuitive and sound (Reviewers w6P6, j25h), the proposed model achieves strong generalization (all reviewers), and the related work is comprehensive and clear (Reviewers w6P6, oDdz).

To address your main concerns, we have done our best to improve our work and revise the paper accordingly. The revised texts in the draft are highlighted in blue. Please refer to the newest draft for the details. The revised parts are also summarized as follows:

1. We put the "Related Work" section right after "Introduction" and add a comparison to previous neural-symbolic methods. (Suggested by Reviewers w6P6 & kopJ)
2. We conduct a machine translation experiment using the English-French translation task from Lake & Baroni (2018), to explore how well the proposed NSR model handles real-world tasks. We put the details on this dataset and results in Appendix, Section A.4, due to the page limit. We also provide a discussion on the potential challenges when applying NSR to real-world tasks in the "Conclusion and Discussion" section. (Suggested by Reviewers w6P6 & j25h)
3. We add an ablation study to analyze the performance of individual components of NSR. We put the results and analysis of this ablation study in Appendix, Section A.5. (Suggested by Reviewer w6P6)
4. We add a comparison to a more recent baseline from Ontanón et al. (2022) in Table 1. (Suggested by Reviewers kopJ & oDdz)
5. We share our code at [bit.ly/nsr-iclr23](https://bit.ly/nsr-iclr23) and will host the code in a GitHub repository and share the trained models and experimental logs in the future.
6. some writing revision. (Suggested by Reviewer oDdz)

We appreciate all the suggestions made by reviewers to improve our work. It is our pleasure to hear your feedback and we look forward to answering your follow-up questions.

Paper264 Authors

---

### Decision · Program_Chairs · 2023-01-20

**Decision:**

Reject

**Justification For Why Not Higher Score:**

The paper proposes a promising approach to improving compositional generalization in neural models, and therefore might be of interest to some researchers, but the approach requires additional experimental verification on non-synthetic tasks.

**Justification For Why Not Lower Score:**

N/A

**Metareview: Summary, Strengths And Weaknesses:**

The paper proposes a neural architecture suitable for learning compositional generalization consisting of 3 modules: perception, dependency parsing and program induction. Since the architecture is not end-to-end differentiable, a sampling-based inference method is proposed. A number of theoretical results regarding the expressiveness and generalization (equivariance and recursiveness) are presented. Experimental results show that the approach performs on par with previous approaches on SCAN (semantic parsing), while outperforming them on PCFG (string manipulation) and HINT (arithmetic reasoning). Despite these results, the main weakness of the paper is that the evaluation is only done on synthetic tasks. Due to the complexity of the proposed approach (consisting of multiple components with the risk of cascading errors), there is a risk that it could be tailored to specific problems while not generalizing beyond those. Therefore it cannot be concluded from the current evaluation that the approach will generalize on tasks with real data.

**Summary Of Ac-Reviewer Meeting:**

We had a virtual meeting regarding this paper attended by 3 reviewers and the area chair. There was agreement that the paper is not strong enough to be accepted. The main issue raised was that the evaluation is only done on synthetic tasks, and due to the complexity of the proposed approach (consisting of multiple components with the risk of cascading errors) it is not clear enough that the approach would generalize on tasks with real data. The perception component was pointed out as a strength of the model, but isn't strong enough as a contribution on its own.